# Molecular Subtypes and Tumor Microenvironment Characteristics of Small-Cell Lung Cancer Associated with Platinum-Resistance

**DOI:** 10.3390/cancers15143568

**Published:** 2023-07-11

**Authors:** Jihyun Kim, Sunshin Kim, Seog-Yun Park, Geon Kook Lee, Kun Young Lim, Jin Young Kim, Jung-Ah Hwang, Namhee Yu, Eun Hye Kang, Mihwa Hwang, Bo Ram Song, Charny Park, Ji-Youn Han

**Affiliations:** 1Research Institute, National Cancer Center, 232 Ilsan-ro, Goyang-si 10408, Kyeonggi-do, Republic of Korea; jihyunk5478@gmail.com (J.K.); ksunshin@ncc.re.kr (S.K.); gklee@ncc.re.kr (G.K.L.); kylim@ncc.re.kr (K.Y.L.); jykim@ncc.re.kr (J.Y.K.); hjungah@ncc.re.kr (J.-A.H.); namheeyu@ncc.re.kr (N.Y.); sidhdgg789@gmail.com (E.H.K.); polaris0628@ncc.re.kr (M.H.); 75758@ncc.re.kr (B.R.S.); 2Korea Disease Control and Prevention Agency, Osong Health Technology Administration Complex, 187, Osongsaengmyeong 2-ro, Osong-eup, Heungdeok-gu, Cheongju-si 28159, Chungcheongbuk-do, Republic of Korea; 3Department of Pathology, National Cancer Center, 232 Ilsan-ro, Goyang-si 10408, Kyeonggi-do, Republic of Korea; 11740@ncc.re.kr

**Keywords:** small-cell lung cancer, molecular subtype, endothelial-to-mesenchymal transition, immune infiltration, platinum resistance

## Abstract

**Simple Summary:**

Although molecular subtypes of small-cell lung cancer (SCLC) have been proposed, their therapeutic implications remain unclear. We dissected SCLC subtypes to delineate the tumor microenvironment (TME) implicated in platinum-drug resistance: ASCL1+ (SCLC-A) subtype of the neuroendocrine type resembled *RB1*/*TP53*-mutant non-SCLC; inflammatory (SCLC-I) subtype presented CD8+/PD-L1+ T-cell infiltration and endothelial-to-mesenchymal transition (EndMT); NEUROD1 (SCLC-N) subtype showed neurotransmission process activation; and POU2F3+ (SCLC-P) subtype showed upregulated epithelial-to-mesenchymal transition (EMT). Meanwhile, the EndMT population was abundant in platinum-resistant SCLC. To overcome platinum resistance, we interrogated drug candidates through high-throughput screening. Cell cycle inhibitors were no longer susceptible to platinum resistance, as opposed to SCLC-A/N. The bromodomain and extra-terminal (BET) inhibitor JQ1 exhibited sensitivity to EndMT promoted by platinum resistance. BET inhibitors are therefore novel therapeutic candidates for overcoming platinum resistance.

**Abstract:**

Although molecular subtypes of small-cell lung cancer (SCLC) have been proposed, their clinical relevance and therapeutic implications are not fully understood. Thus, we aimed to refine molecular subtypes and to uncover therapeutic targets. We classified the subtypes based on gene expression (*n* = 81) and validated them in our samples (*n* = 87). Non-SCLC samples were compared with SCLC subtypes to identify the early development stage of SCLC. Single-cell transcriptome analysis was applied to dissect the TME of bulk samples. Finally, to overcome platinum resistance, we performed drug screening of patient-derived cells and cell lines. Four subtypes were identified: the ASCL1+ (SCLC-A) subtype identified as *TP53*/*RB*-mutated non-SCLC representing the early development stage of SCLC; the immune activation (SCLC-I) subtype, showing high CD8+/PD-L1+ T-cell infiltration and endothelial-to-mesenchymal transition (EndMT); the NEUROD1 (SCLC-N) subtype, which showed neurotransmission process; and the POU2F3+ (SCLC-P) subtype with epithelial-to-mesenchymal transition (EMT). EndMT was associated with the worst prognosis. While SCLC-A/N exhibited platinum sensitivity, the EndMT signal of SCLC-I conferred platinum resistance. A BET inhibitor suppressed the aggressive angiogenesis phenotype of SCLC-I. We revealed that EndMT development contributed to a poor outcome in SCLC-I. Moreover, heterogenous TME development facilitated platinum resistance. BET inhibitors are novel candidates for overcoming platinum resistance.

## 1. Introduction

Small-cell lung cancer (SCLC) accounts for approximately 15% of all lung cancers and has an extremely poor prognosis [1]. The standard chemotherapy involves cisplatin and etoposide, which has remained unchanged over the past three decades. Despite initial sensitivity to chemotherapy, most patients develop early recurrence, resulting in a poor prognosis. Although anti-PD-1 immunotherapy improved survival, the benefits were modest [2,3,4]. Unfortunately, most patients ultimately develop disease progression within 2 years [2]. Therefore, novel therapeutics that will help prolong survival are urgently required.

SCLC molecular profile uncovered the genomic characteristics of disease progression from de novo development to relapse [1]. Lung adenocarcinoma (LUAD) patients receiving epidermal growth factor receptor (EGFR) tyrosine kinase inhibitor (TKI) therapies undergo SCLC neuroendocrine transformation with *RB1*/*TP53* mutations as well as enhancement of the PRC2 complex and PI3K/AKT pathways [5]. The SCLC driver mutations *RB1*/*TP53* facilitate cell cycle progression in tumorigenesis, and limited-stage SCLC is activated through the PI3K/AKT pathway [1,6]. In furtherance of a bulk sample biopsy, a single-cell transcriptome dissected the heterogeneity of tumor cells and tumor microenvironment (TME) cells. Platinum resistance facilitates development of the TME and promotes progression to the inflammatory type [4]. Following immune cell infiltration, pro-fibrotic macrophages promote metastatic SCLC and its poor prognosis [7]. Elucidation of gene expression in TME cells could be utilized to devise potential therapeutic strategies. However, clinical trial data on molecular features upon relapse are insufficient, and cell lines used in such studies have limitations in terms of reflecting the patient response to the drugs. Thus, the efforts to treat SCLC have not achieved remarkable success.

Pharmacogenomics platforms have been used to identify new drug candidates and develop new therapeutics, and their molecular mechanisms have been studied using high-throughput screening [8,9]. However, such platforms have several limitations. Pharmacogenomic platforms are restricted to in vitro models, and preclinical studies employing mouse models are limited to few candidates [8]. Furthermore, molecular profiles obtained from bulk samples could help detect only a large proportion of target cells and were unable to identify rare cell populations or continuous transitions [10]. Consequently, resistance to existing therapies is not fully understood [9,11]. To overcome these limitations, single-cell level analysis could uncover rare cell regulatory mechanisms and resistant transitions [4,7]. Identifying elements involved in therapeutic resistance could help develop breakthrough therapies that will simultaneously target both tumors and drug-tolerant subpopulations [10]. However, data obtained from systematic investigations are inadequate in the single-cell level resistant population and its target drugs for SCLC.

In this study, we classified SCLC molecular subtypes using gene expression profiling of patient biopsy samples of two cohorts: our National Cancer Center cohort (NCC; *n* = 87) and the George et al. cohort (*n* = 81) [12]. To define the solid subtype, we examined the regulatory mechanism, clinical features, and pathological diagnosis by comparing the two cohorts. To investigate the early developmental status, the SCLC expression profile was compared with that of non-SCLC (NSCLC) samples harboring *TP53* and *RB1* mutations. Subsequently, we obtained a single-cell transcriptome profile to dissect targetable cell populations [7]. Tumor and microenvironment cell populations were classified, and therapy-resistant cells were segregated into subpopulations. The abundance of cell subpopulations was compared for each molecular subtype and prognosis based on cellular regulation. Finally, platinum resistance-related gene sets were extracted from cell subpopulations, and drugs eliciting sensitivity were identified using data from drug screening performed on patient-derived cells.

## 2. Materials and Methods

### 2.1. Patient Enrollment and Sample Collection for the NCC Cohort

We retrospectively reviewed the medical records and collected tumor samples from 172 consecutive patients with SCLC who were treated between March 2001 and April 2018 at the NCC Hospital in Korea. Two pathologists (S.Y.P. and G.K.L.) confirmed the histology of SCLC and assessed the tumor cell content within each tumor sample. Tissues with >20% tumor cells in hematoxylin-and-eosin-stained sections were subjected to NanoString analysis and multiplex immunohistochemistry (mIHC). Tumor-node-metastasis (TNM) staging was based on the 8th edition of the lung cancer TNM staging system and was confirmed by a radiologist (K.Y.L.).

### 2.2. Gene Expression Analysis of SCLC to Identify Molecular Subtypes

To define the SCLC molecular subtypes, we clustered the gene expression profiles acquired from patient RNA-seq samples from the George et al. dataset (*n* = 81) [12]. Cluster sizes of 3–5 were investigated and optimized to 4 using silhouette plots via the non-negative matrix factorization (NMF) method (Figure 1A). We extracted 3,604 upregulated DEGs using the R package limma (*p* < 0.01) for each subtype [13]. Subsequently, to investigate the role/activities of each subtype within the pathway, we performed gene set variation analysis (GSVA) using the expression profile and DEG sets [14,15]. Additionally, the molecular subtypes of SCLC cell lines (*n* = 50) were equally classified using NMF clustering [16].

The gene expression profile of the NCC cohort was assessed using the NanoString PanCancer Panel from formalin-fixed paraffin-embedded (FFPE) samples. A panel assay was used to quantify the expression of 730 genes. We used the R package NanoStringNorm and performed normalization procedures by adjusting for housekeeping genes and background correction following the recommendations of NanoString [17]. After preprocessing, we excluded samples with missing information for more than 90% of the endogenous genes or those with high standard deviations (>3). We also clustered samples using NMF. Because the panel did not contain *ASCL1*, *NEUROD1*, or *POU2F3*, four subtypes were identified according to the scores of the three master regulator target genes using GSVA [15]. Target gene signatures were acquired from lung cancer chromatin immunoprecipitation sequencing (ChIP-seq) studies or known gene set signatures of ASCL1-target, NEURO-act, and POU2F3-target [18,19,20]. Additionally, we calculated NE scores according to a previously reported method [8,21].

### 2.3. Gene Expression Comparison between TP53/RB1-Mutated NSCLC and SCLC

Inactivation of TP53 and RB1 are key drivers of SCLC, and these variants have already been observed in NSCLC [5,12]. Particularly, *TP53/RB1* inactivation in NSCLC (NSCLC*^TP53/RB1^*) could be considered a predictor of histological transformation into SCLC [5]. Therefore, we assessed the molecular characteristics of NSCLC*^TP53/RB1^* compared with SCLC subtypes. We selected NSCLC*^TP53/RB1^* samples from TCGA LUAD and LUSC datasets [22,23]. Strict conditions were applied to select inactivation mutations (indels, splicing site, and nonsense), and no sample was found to contain RNA fusions in either TP53 or RB1. We divided TCGA lung cancer samples into two groups, namely, NSCLC*^TP53/RB1^* (MUT) and wild type (WT), and performed DEG analysis to identify SCLC master regulators. Additionally, we compared the expression profiles of LUAD*^TP53/RB1^* samples with those of the SCLC samples. Before comparison, we eliminated batch effects between TCGA and George expression profiles using Combat and performed principal component analysis (PCA) for all samples [24]. To evaluate the NE characteristics, we calculated the NE scores according to a previously described method [21]. Cox regression analysis of PFS using *TP53/RB1* LUAD samples was also performed.

### 2.4. Single-Cell Transcriptome Analysis to Dissect the SCLC TME

The single-cell transcriptome profile of malignant SCLC cells was obtained from the Human Tumor Atlas Network [7]. We retrieved single-cell transcriptomes from seven treatment-naïve and ten drug-treated SCLC patients, combined cells for each treatment status group, and assessed the unique molecular identified (UMI) count groups with the following thresholds: (1) zero UMI counts for 80% of all genes, and (2) the standard deviation of all genes per cell lower than 1. The combined matrix contained 6585 treatment-naïve cells and 5927 drug-treated cells. The UMI counts for each gene were normalized to the total UMI count per cell and transformed into a log scale using the R package Seurat [25]. Subsequently, we distinguished neuroendocrine (NE) and non-NE cells by calculating NE scores through the application of a previously reported method for each cell based on Spearman correlation analysis using a 50-gene signature [26]. We computed two-dimensional data using t-distributed stochastic neighbor embedding (t-SNE). A likelihood ratio test based on Poisson distribution was performed to identify DEGs within each cluster from the t-SNE analysis. Finally, each cell cluster was annotated using known cell marker expression. To compare the functional characteristics of the cell types according to the NE score, we performed a limma test and GSVA pathway analysis. The clonal trajectory tree was inferred using the R package Monocle orderCells [27].

### 2.5. Survival Analysis to Identify Molecular Features Related to SCLC Outcomes

To test the survival significance of OS and PFS, Cox regression analysis was performed for cancer stage (LD and ED) and NE score (low and high). We also performed GSVA to determine cell distribution in the bulk RNA-seq dataset based on cell markers identified from single-cell analysis, including cisplatin resistance markers, from the mSigDB drug signature database [28,29]. To classify samples according to the five TME cluster types, we estimated GSVA scores for each TME cell cluster and subsequently assigned TME subcluster types corresponding to the maximum score for each sample.

### 2.6. IHC

Consecutive four-micrometer-thick tissue sections were cut from FFPE tissues for IHC. IHC staining was performed using a Ventana automatic immunostainer (Ventana, Benchmark XT, Tucson, AZ, USA) following standard automated protocols. Ventana Retrieval Solution CCl (equivalent to EDTA buffer, pH 8.0) was used for epitope retrieval for 30 min. The primary antibodies (ASCL1:1:100, Clone EPR19840, Abcam, Cambridge, UK; NEUROD1:1:4000; Clone IMR-32, Abcam, Cambridge, UK; POU2F3:1:50, polyclonal, NBP1-83966, Novus Biologicals, Centennial, CO, USA) were incubated for 32 min at 42 °C and detected using the Ultra-View Detection Kit (Ventana) with DAB as the chromogen. FFPE cell line pellets with known protein expression of ASCL1, NEUROD1, and POU2F3 were used to establish optimal IHC conditions and assess the sensitivity and specificity of each antibody. The proportion of positive tumor cells was calculated as described previously [30].

### 2.7. Multiplex Immunofluorescence (mIF) Staining and Analysis of the Immune Cell Population

mIF staining and analysis were performed using a Leica Bond Rx autostainer (Leica Microsystems, Buffalo Grove, IL, USA). After adding anti-CD8 primary antibody (MCA1817, Bio-Rad, Hercules, CA, USA, dilution 1:300), the slides were incubated for 30 min. The polymer HRP Ms+Rb secondary antibody (ARH1001EA, Akoya Biosciences, Marlborough, MA, USA) was applied for 10 min for detection. Staining for CD8 was visualized using Opal 570 TSA Plus (dilution 1:150) for 10 min. Subsequently, the slides were treated with Bond Epitope Retrieval 1 (#AR9961, Leica Biosystems) for 20 min to eliminate bound antibodies before the next step. In a serial fashion, anti-CD4 (ab133616, Abcam, dilution 1:200), anti-PD-L1 (13684S, CST, dilution 1:300), anti-FOXP3 (ab20034, Abcam, dilution 1:100), anti-CD20 (ab9475, Abcam, dilution 1:100), and anti-CK (NBP2-29429, NOVUS, dilution 1:300) antibodies were applied for 30 min, and the polymer HRP Ms+Rb secondary antibody (ARH1001EA, Akoya Biosciences, Marlborough, MA, USA) was applied for 10 min for detection. Staining of the five proteins was visualized using Opal 520 TSA Plus for 10 min. The slides were then treated with Bond Epitope Retrieval 1 (#AR9961, Leica Biosystems) for 20 min to remove the bound antibodies before the next step for each condition. The nuclei were subsequently visualized using DAPI, and coverslips were mounted on each slide with ProLong Gold antifade reagent (P36934, Invitrogen, Waltham, MA, USA).

The slides were scanned using the Vectra Polaris Automated Quantitative Pathology Imaging System (Akoya Biosciences, Marlborough, MA, USA), and images were analyzed using InForm 2.2 software (Akoya Biosciences) and TIBCO Spotfire (TIBCO Software, Palo Alto, CA, USA). To acquire reliable unmixed images, representative slides of each emission spectrum and unstained tissue were scanned. Each stained section was used to establish the spectral library of fluorophores required for multispectral analysis, which included the reference for target quantitation, as the intensity of each fluorescent target was extracted from the multispectral data using linear unmixing. Each cell was identified by detecting the nuclear spectral elements (DAPI). All immune cell populations from each panel were characterized and quantified using the cell segmentation tool of InForm image analysis software. For co-expression analysis, the data obtained from InForm were uploaded to Spotfire software, and the threshold for positivity of each antibody was determined based on IHC scoring methods.

### 2.8. Drug Response Screening of PDCs

PDCs were collected from patients with advanced or refractory lung cancer treated at the National Cancer Center, Korea, between December 2016 and February 2022. The histological type of the lung tumor was based on the 2004 World Health Organization (WHO) classification. This study was approved by the National Cancer Center Institutional Review Board (Approval No.: NCC2016-0208). Written informed consent was obtained from all patients. PDCs were acquired from pleural effusion, pericardial effusion, ascites, and tissues. PDCs were isolated from liquid samples (pleural effusions) using density gradient centrifugation and from tissue samples using fine mincing. After washing, the cells were cultured in AR-5 medium (5% FBS, 1X GlutaMAX (Thermo Fisher Scientific, Waltham, MA, USA), 1X Insulin-Transferrin-Selenium (ITS, Thermo Fisher Scientific), 1% penicillin–streptomycin, 50 nM hydrocortisone, 1 mM sodium pyruvate, and 1 ng/mL epidermal growth factor (EGF) in RPMI 1640) at 37 °C in a 5% CO_2_ atmosphere. The medium was changed carefully every 2–3 days until the cells stabilized in the flask. Stabilized PDCs were seeded in 384-well plates (1000 cells/20 μL/well) in quadruplicate for each treatment. A total of 64 compounds were used to screen each PDC sample. After overnight incubation, the cells were treated with one drug at a five-fold serial dilution for six doses (50–16 nM). Cell viability was measured after 72 h of treatment using the CellTiter-Glo Luminescent Cell Viability Assay Kit (Promega Corporation, Madison, WI, USA). Each screening plate contained a dimethyl sulfoxide (DMSO)-only vehicle to calculate relative cell viability and normalize the data. Dose-response curve (DRC) fitting and area under the curve (AUC) values were assessed using GraphPad Prism 5.3 (GraphPad Software Inc., San Diego, CA, USA). All the compounds were purchased from Selleckchem (Houston, TX, USA).

### 2.9. Drug Susceptibility Investigation for SCLC Cell Lines and PDCs

To assess the drug sensitivity of SCLC cell lines (*n* = 50) from the CCLE database, we calculated the NE scores and defined the cell types from markers of single-cell analysis, as described above. Subsequently, we tested drug sensitivity by t-test using rescaling response values [$-log({IC}_{50})$] for NE (*n* = 40) vs. non-NE (*n* = 10) or endothelial cell type (*n* = 10) vs. other type (*n* = 40) samples. Next, we calculated the correlation of the rescaled response values of JQ1 with the expression of each gene in SCLC cell lines. Gene set enrichment analysis (GSEA) was performed using highly correlated genes.

To explore drug candidates from the screening profile of 65 drugs and 366 PDCs (345 NSCLC and 21 SCLC), we calculated the FC in the half-maximal inhibitory concentration (IC_50_) value between NSCLC and SCLC or between cisplatin-sensitive and cisplatin-resistant SCLC cells. Differences between the two groups were tested using the t-test. The samples were classified into two cisplatin response groups based on the cisplatin IC_50_ values.

## 3. Results

### 3.1. Molecular Subtypes of SCLC and Their Clinical Relevance

In the respective analysis of the two patient cohorts and cell lines, SCLC was classified into four subtypes: SCLC-A of *ASCL1*^+^, SCLC-I of *CD274*^+^ (PD-L1), SCLC-N of *NEUROD1*^+^, and SCLC-P of *POU2F3*^+^ (Figure 1A,B). We verified the targeted panel expression profile of the NCC cohort and compared it with that of the whole transcriptome in the George et al. dataset [12]. The two datasets exhibited high correlation coefficients with the four subtypes (*R* > 0.73; Figure 1A). Three master regulators, *ASCL1*, *NEUROD1*, and *POU2F3,* were upregulated in the three subtypes SCLC-A/N/P (Figure 1B and Appendix A). SCLC-I was observed more frequently in the NCC cohort (29.9%) than in the George et al. cohort (17.3%) [12], whereas SCLC-N showed the opposite trend (NCC, 14.9%; George et al. 33.3% [12]; Appendix A). In terms of clinical profiles, the NCC cohort comprised 80.5% of ED (stage IV) cases, and the George et al. cohort comprised 55.6% of ED cases [12]. In terms of the clinical profile of NCC, the SCLC-A group was the youngest, and the SCLC-P group was the oldest (*p* = 0.01; Figure 1C and Appendix A). Brain metastasis (NCC, 14.9%) was more frequently observed in the SCLC-I (OR = 1.57) and SCLC-P (OR = 1.71; Figure 1C) groups.

Pathological investigation of the master regulators further validated the performance of our molecular subtype classification (Figure 1C). We performed immunohistochemical (IHC) analysis for various markers and assessed regulation of the signature target genes bound by the three master regulators. The IHC showed that ASCL1^+^ was enriched in SCLC-A and SCLC-I/N. POU2F3 clearly defined only the SCLC-P subtype (SCLC-P, *p* = 3.89 × 10^−4^; Figure 1C). Moreover, NEUROD1 IHC staining was non-specific and failed to classify SCLC-N. Additionally, TTF1^+^ was relatively abundant, except in SCLC-P. MYC^+^ was abundant in SCLC-I and SCLC-P. CD56 was a positive marker for all SCLC samples.

Differentially expressed genes (DEGs) were used to define the distinct regulatory programs for each subtype. The expression of known marker genes was consistent with the expected results (*p* < 6.0 × 10^−5^; Appendix A). *ASCL1*, *DLL3*, *FOXA1*, and *SOX2* were overexpressed in SCLC-A cells. Wnt signaling and FOXA1/3 transcription factor network pathways were enriched in SCLC-A (Figure 1D, Appendix A). *CD274* (PD-L1) and T-cell receptor (TCR) signaling was activated in SCLC-I. High NE scores were observed for both SCLC-A and SCLC-N. The mitotic cell cycle was dysregulated in both SCLC-A/N cell lines. PI3K-Akt signaling was activated in SCLC-N. *POU2F3* and *MYC* were dramatically upregulated in SCLC-P cells. Notch signaling and extracellular matrix (ECM) organization contributed to the SCLC-P regulatory program.

### 3.2. Early Developmental Status of SCLC Identified from TP53/RB1-Mutated NSCLC Resembles That of SCLC-A

Both *TP53* and *RB1* variants are key drivers of SCLC development [5,12]. The histological transformation from NSCLC to SCLC confers therapeutic challenges in sharing *TP53*/*RB1* variants [5]. To define the early developmental status of SCLC in NSCLC, we explored whether the transcriptome profile of NSCLC accompanies the subtype features of SCLC. Accordingly, we investigated the gene expression profiles of samples harboring *TP53*/*RB1* variants from The Cancer Genome Atlas (TCGA) LUAD (*n* = 510) and squamous cell carcinoma (lung squamous cell carcinoma (LUSC); *n* = 484) datasets. We extracted data for the LUAD*^TP53^*^/*RB1*^ (*n* = 4) and LUSC*^TP53^*^/*RB1*^ (*n* = 7) samples.

*ASCL1* was significantly upregulated in LUAD*^TP53/RB1^* samples compared with that in wild-type LUADs (*p =* 0.02); however, no difference was observed in *NEUROD1* (*p =* 0.5) and *POU2F3* (*p =* 0.83; Figure 2A). In LUSC, *NEUROD1* was upregulated in LUSC *^TP53/RB1^* (*p =* 0.05, Figure 2A). NE scores were significantly higher in LUAD*^TP53/RB^* (*p* = 0.0004) than wild-type LUAD, but not in LUSC*^TP53/RB^* (*p* = 0.59; Figure 2B). We also analyzed the expression of SCLC DEG regulators in LUAD. The SCLC-A regulators *FOXA1* and *SOX2* were upregulated in LUAD*^TP53/RB^* (*p* < 0.003). In contrast, the genes of the remaining three subtypes SCLC-I/N/P (*CD274*, *MYC*, *NOTCH2*, *REST*, and *SOX9*) showed no difference in expression (Figure 2C). Upon comparing the global gene expression and NE scores of LUAD *^TP53/RB^* with those of SCLC subtypes, LUAD *^TP53/RB^* expression was found to be close to that for SCLC-A and SCLC-N (Figure 2D,E). Additionally, LUAD*^TP53/RB^* showed a trend toward worse progression-free survival (PFS) (hazard ratio [HR] = 2.65; *p =* 0.08) and overall survival (OS) (*p =* 0.058). Based on these findings, we concluded that LUAD*^TP53/RB^* exhibits SCLC NE gene expression, and its molecular features are similar to those of the SCLC-A subtype.

### 3.3. Single-Cell Clusters to Dissect Endothelial-to-Mesenchymal Transition (EndMT) Phenotypes in SCLC-I Type

Although NE-type cell development is well characterized, the SCLC TME is not well defined. A previous study involving immune cell analysis revealed cytotoxic T-cell infiltration in SCLC-I and a dysregulated immune response by exhausted T cells in SCLC-N [7]. However, the TME cell types and their evolution are not fully understood. Therefore, we next investigated the single-cell transcriptome to select non-NE cells to identify TME cells. We also examined TME cell subclusters and clonal evolution. To characterize TME cells, we acquired a single-cell transcriptome profile of patient malignant cells (treatment-naïve: 7 patients, 6585 cells; previously treated: 10 patients, 5927 cells; as described in Materials and Methods) [7]. Previously treated patients were administered platinum-based cytotoxic agents (etoposide, cisplatin, or carboplatin) and immune checkpoint inhibitors (ipilimumab or atezolizumab).

We classified NE and non-NE cells and clustered the SCLC TME cells with non-NE cells. *POU2F3*^+^ staining was observed among the subclusters of non-NE cells. Interestingly, post-treatment SCLC cells comprised 14.2% non-NE cells, whereas treatment-naïve cells comprised 1.1% non-NE cells (Figure 3A). Therefore, we selected non-NE cells from the post-treatment samples. Compared with NE cells, which exhibit *ASCL1* and *TFF3* upregulation (*p* < 1.42 × 10^−111^), heterogenous non-NE cells presented *IFITM3*, *B2M*, *ANXA4*, *VIM*, *CD74*, *S100A11*, and *YAP1* overexpression (*p* < 5.77 × 10^−57^, fold change (FC) > 1; Appendix A). Interferon signaling, cell cycle, and antigen processing were simultaneously activated in non-NE cells (Appendix A). NE cells exhibited *ASCL1*, *TFF3*, and *DLL3* upregulation (Appendix A).

Non-NE cells, mostly TME cells, were extracted from all previously treated patients without any sample bias (Appendix A). Subsequently, non-NE cells were classified into five clusters (Figure 3B and Appendix A). Five clusters were annotated using overexpression markers: low-NE, myeloid, epithelial, T-cell, and endothelial (Figure 3B; *p* < 1.27× 10^−5^; Appendix A; Appendix A). ASCL1 was relatively active in low-NE cells to maintain cell cycle activation and P53 signal dysregulation (*p* < 6.18 × 10^−4^; Appendix A; Appendix A). *POU2F3* upregulation was observed in the epithelial cluster (*p* = 4.62 × 10^−15^; Appendix A). Myeloid cells also exhibited *MKI67*, *CDK1*, and *TOP2A* overexpression, contributing to an aberrant cell cycle (Appendix A). The T-cell cluster was identified using *PTPRC* (*CD45*) staining. Additional T-cell subtypes were evaluated using IHC on the NCC cohort samples. IFN-γ and TCR staining were detected for T-cell subtype genes. Furthermore, the endothelial clusters were found to modulate the ECM, angiogenesis, and vascular endothelial growth factor (VEGF) signaling. The epithelial cluster showed enrichment in the biological processes of cell adhesion, interferon-alpha/beta, and TGF-beta signaling. These results suggest that *VIM*^+^ cancer-associated fibroblast (CAF) features govern endothelial and epithelial clusters. These two clusters were considered the previously defined EndMT and *POU2F3*^+^ epithelial-to-mesenchymal transition (EMT) clusters, respectively.

The non-NE SCLC population evolved into distinct TME cell types. The five TME cell clusters expanded from low-NE cells to three distinct branches: myeloid, T, and endo/epithelial cells (Figure 3C). We further investigated the abundance of the five TME cell types among the SCLC subtypes. Low-NE cells were enriched in SCLC-A (*p* = 2.28 × 10^−7^; Figure 3D). Endothelial and T-cell signatures were abundant in SCLC-I (*p* < 7.0 × 10^−4^). The remaining cell types were not significantly different among the four SCLC subtypes. Moreover, we verified T-cell infiltration using an mIHC/IF-based assay in the NCC cohort (Figure 3E and Appendix A). CD4^+^ and CD8^+^ T cells were abundant in SCLC-I, as were CD4^+^/FOXP3, CD8^+^/PD-L1, and CD20 (*p* < 0.06; S18-26664; Kruskal–Wallis H test, *p* < 0.004; Figure 3F and Appendix A). Non-NE cells showed an apparent increase in number after treatment and evolved into three branches: myeloid, T, and endo/epithelial cells. Specifically, SCLC-I comprised a heterogeneous TME to facilitate EndMT and CD8^+^/PD-L1 T-cell infiltration.

### 3.4. SCLC TME Cell Cluster Signatures to Predict the Worst Outcome

Prognostic features of SCLC have been demonstrated in both restricted cohorts and mouse models. Here, we referred to two cohorts to explore the genetic regulations promoting poor outcomes. First, ED patients were found to have poor survival compared with limited disease (LD) patients in terms of both 2-year PFS (*p* = 0.07, HR = 1.85) and OS (*p* = 0.01, HR = 2.65) in NCC (Figure 4A). The PFS and OS according to LD/ED status were not significant in the George et al. cohort [12]. However, the non-NE-type clearly distinguished poor outcomes between the two cohorts (PFS George et al. [12], *p* = 0.09; NCC, *p* = 0.1; Figure 4B).

Additionally, we performed survival tests for TME molecular features (Appendix A). Survival, according to the four SCLC subtypes, failed to reach statistical significance. However, the regulatory programs reflected by the five TME subclusters were prognostic factors. In the George et al. cohort, T-cell infiltration was associated with a better prognosis (Figure 4C). Compared with patients with T-cell activation, endothelial signal-activated patients exhibited worse outcomes (*p* = 0.1, HR = 1.84). CD4, CD8, and T_reg_ IHC staining for the NCC cohort was similar to that for the George et al. cohort (Appendix A). Because only genes of three signatures were available in the NCC-targeted panel, we referred to myeloid signaling for comparison. An endothelial signature indicated the worst outcome (*p* = 0.14, HR = 1.73; Figure 4C, bottom panel). We presumed that the TME heterogeneity of SCLC-I encompasses contradictory prognostic factors. Even with T-cell infiltration, the endothelial signature indicated the worst outcome in the SCLC-I subtype.

### 3.5. SCLC Molecular Features and Drug Candidates for Resistance to Platinum-Based Chemotherapy

To characterize the initial sensitivity and rapid progression of SCLC to chemotherapy, we referred to the cisplatin response scores analyzed from the SCLC transcriptome profile. SCLC-A and SCLC-N were sensitive, whereas SCLC-I was the most resistant (*p* = 6.84 × 10^−6^; Figure 5A) [28]. Among the TME cell types, the endothelial and T-cell-activated groups also participated in cisplatin resistance (*p* = 1.79 × 10^−7^; Figure 5A). This implied that TME development in the SCLC-I subtype was associated with cisplatin resistance, whereas the two NE subtypes remained sensitive to cisplatin.

We evaluated cisplatin resistance scores from high-throughput drug screening data for cell lines and PDCs. The drug response values of the SCLC cell lines also resembled cisplatin response scores (Figure 5B). Unfortunately, the IC_50_ values of cisplatin and etoposide are unavailable for the Cancer Cell Line Encyclopedia (CCLE) [16]. However, the NE-type cells were consistently sensitive to platinum-based drugs (oxaliplatin; FC = 0.34, *p* = 0.04) and histone deacetylase (HDAC) inhibitors (MS-275 and SB939; FC > 0.28, *p* < 0.09). Moreover, the endothelial-activated cell types were sensitive to bromodomain and extra-terminal motif (BET) inhibitors (JQ1 and GSK-1210151A; FC > 0.55, *p* < 0.05) and a graft inhibitor (pelitrexol; FC = 0.43, *p* < 0.08). To cross-evaluate the CCLE results, we investigated additional drug response screening profiles acquired for 366 PDCs and 64 drugs (Appendix A). Upon comparing SCLC (*n* = 21) with NSCLC (*n* = 341), drugs targeting Chk1, Aurora, Wee1, and PI3K-Akt were identified as candidates for SCLC sensitivity (Figure 5C). However, the comparison between cisplatin-sensitive and -resistant groups within SCLC indicated different candidates compared with NSCLC. Cell-cycle target drugs were no longer predicted to be effective against cisplatin-resistant SCLC PDCs. Cisplatin-resistant PDCs exhibited concurrent resistance to barasertib, alisertib (Aurora), camptothecin (CPT; topoisomerase), and everolimus (mTOR; FC < −1.26, *p* < 0.04). Effective drugs for cisplatin-resistant cells were barely identified with a strict *p* value cutoff. Brigatinib (ALK/EGFR inhibitor), dasatinib (multiple TKI), and JQ1 (BET inhibitor) were considered potential candidates based on FC (FC > 0.61), even though the *p* value cutoff was not reached. Our results imply that the response to chemotherapy resembled that of cell cycle-targeting drugs or other cytotoxic drugs.

Finally, we performed an integrative analysis of the co-expressed regulatory target genes based on the candidate JQ1 response. Upon assessing highly correlated genes affected by JQ1 treatment (negative IC_50_), we observed enrichment of the extracellular matrix organization, Hippo, WNT, and angiogenesis pathways (*p* < 3.85 × 10^−3^, Appendix A). *CCND1* was the top-ranked factor involved in the Hippo and WNT signaling pathways (*R* = 0.7, *p* = 9.94 × 10^−8^; Figure 5D). *TGIF2* and *SMAD3* in the TGF-β signaling pathway were also highly correlated (*R* > 0.59, *p* < 0.002). Other correlated pathways were also associated with EndMT. JQ1, a BET inhibitor targeting BRD4, suppresses angiogenesis in various cancer types [31]. As reported previously, EndMT in non-NE cell types increases platinum resistance. To overcome platinum resistance, our data propose BET inhibitors as novel therapeutic candidates for suppressing angiogenesis activation in SCLC-I.

## 4. Discussion

In this study, we scrutinized four molecular subtypes of SCLC. Our pathological investigation suggested CD56, MYC, and TTF1 as positive markers in addition to the three master regulators. SCLC-I was enriched in CD8^+^/PD-L1^+^ cells with relatively low NE expression. The rate of brain metastasis for SCLC is 10% at diagnosis and 40–50% after progression [1]. In the present study, brain metastasis was implicated in non-NE-type SCLC-I and SCLC-P. The histological transformation from NSCLC to SCLC increases the therapeutic challenges in overcoming EGFR TKI resistance in LUAD; however, the molecular features modulated during transformation remain unclear [5]. In a previous study, NE transformation from LUAD was accompanied by activation of the PI3K/AKT and cell cycle pathways [32]. In addition, our study found that LUAD was transformed into the SCLC-A type. In LUAD*^TP53/RB1^*, *SOX2* and *FOXA1* were concurrently upregulated along with *ASCL1*. Based on these findings, we inferred that SCLC-A represents an early developmental state among the SCLC subtypes. Therefore, we propose that appropriate treatments, such as those involving cell-cycle inhibitors or platinum-based cytotoxic agents, can target early NE-type cells evolving from NSCLC and prevent the histological transformation from LUAD to SCLC [33].

Our results highlight the role of EndMT and its angiogenic processes in SCLC-I. SCLC-I was previously identified as an inflamed subtype with infiltrated cytotoxic T cells, and immunotherapy has been shown to improve survival [4]. EMT activation and platinum resistance have also been observed in SCLC-I cells. Furthermore, single-cell transcriptome analysis revealed detailed non-NE populations of dysregulated T cells in SCLC-N and the myeloid milieu associated with idiopathic pulmonary fibrosis [7]. Furthermore, our results dissected the CAF-like fibrotic cell populations into subclusters. Regulation of angiogenesis by EndMT was distinct from that by EMT and was associated with the worst survival outcome. As shown by accumulating single-cell level CAF studies, lung CAFs are classified into subtypes [34]. Among them, EMT and EndMT subtypes exhibit similar molecular characteristics and share the TGF-β pathway regulated by Slug, Snail, Zeb1/2, and Twist [35]. EndMT is a distinct biological process governed by sprouting angiogenesis, whereas EMT is involved in epithelial tubulogenesis [34,35,36]. We expect that the upregulation of EndMT genes according to SCLC therapeutic evolution can be evaluated in expanded cohorts and mouse models.

Moreover, SCLC-P, comprising approximately 15% of SCLC cases, originates partially from tuft cell-like cells [20]. The subtype exhibited distinct biological functions governed by *SOX9* and *MYC* and was associated with a significantly higher rate of ED stage and brain metastasis. However, only one patient was identified as having the SCLC-P type in the single-cell profile analysis [7]. The global expression and standard deviation of *POU2F3* were extremely low in the single-cell profile; therefore, we used the target gene signature acquired from ChIP-seq. We found that *POU2F3*^+^ type cells among the TME subclusters were restricted to the epithelial-type population. However, the epithelial gene signature did not represent a significantly strong signal in the bulk SCLC-P-type samples. Only T-cell abundance was deterministic in distinguishing SCLC-I and SCLC-P from the five TME subclusters. An accumulation of biopsy samples is required to evaluate the single-cell-level transcriptome to delineate the regulatory program of POU2F3 and its tumorigenic role in SCLC.

Resistance to chemotherapy is a critical challenge for improving SCLC therapeutics. Previously developed drugs to inhibit Aurora, Chk1, ATR, Wee1, and topoisomerase only elicited early response for the NE-type, resembling the effects of platinum-based chemotherapy, cisplatin, and etoposide [1]. Our patient-derived tumor cell (PDC) screening results suggested that the resistance to these drugs was acquired simultaneously, and the appropriate therapeutic candidates for platinum-resistant SCLC were challenging to identify. CAF development to activate TGF-β, EMT, and angiogenesis disturbed the platinum response. Despite their low statistical significance, dasatinib (an ATP-competitive TKI) and JQ1 (a BET inhibitor) were effective against platinum-resistant cells. Integrative analysis of the JQ1 response and gene expression profile revealed inhibition of Hippo, TGF-β, and angiogenesis. In a previous study, JQ1 reduced TNBC spheroid growth. The model was more physiologically revenant, as it created oxygen, nutrient, and pH gradients to imply hypoxia and angiogenesis [37]. Meanwhile, SCLC NE cells also exhibited JQ1 sensitivity [38,39]. These findings implied that JQ1 could become a therapeutic candidate to target platinum-resistant SCLC facilitating angiogenesis. We expect that a combination of EndMT-targeting drugs could overcome platinum resistance.

Our study denoted a SCLC regulatory mechanism and therapeutic candidates according to four molecular subtypes. In particular, the platinum resistance of SCLC is a clinical challenge not yet resolved. Our pharmacogenomic analysis highlighted a BET inhibitor to target EndMT promoted by cisplatin resistance. Despite comprehensive analysis to delineate a therapeutic mechanism, a functional assay was insufficient to evaluate the JQ1 target epigenetic mechanism. In future studies, we expect therapeutic mechanism validation to overcome platinum-resistant SCLC.

## 5. Conclusions

Our SCLC subtype analysis identified heterogeneity in the cellular and clinical characteristics. Particularly, analysis of TME subpopulations revealed that EndMT is abundant in SCLC-I and leads to the worst possible outcome. In terms of therapy, the molecular features of the subtypes indicated that BET inhibitors could effectively combat standard chemotherapy resistance.

## Figures and Tables

**Figure 1 cancers-15-03568-f001:**
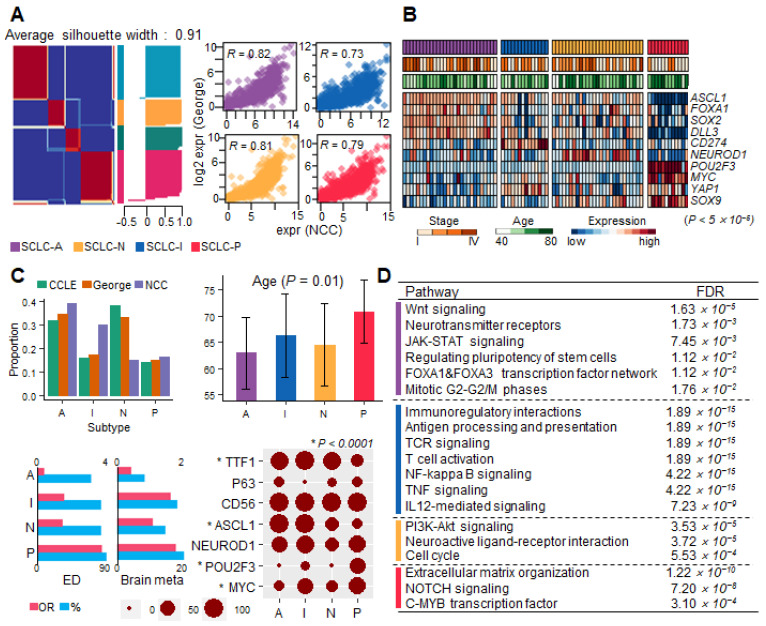
SCLC molecular subtype classification. The four subtypes are denoted by color (SCLC-A, purple; SCLC-N, yellow; SCLC-I, blue; and SCLC-P, red). (**A**) Gene expression profile classification of the two cohorts. The silhouette plot shows that NMF clustering was optimized to cluster size 4. The four scatter plots for each subtype show a high correlation between the expression profiles of the NCC cohort (*x*-axis) and George et al.’s cohort (*y*-axis) [12]. (**B**) Differential expression of known master regulators (*p* < 5.0 × 10^−5^). Subtype, tumor stage, and age are presented on top. (**C**) Clinical profile summary of subtypes. Bar plots present proportions and average age (with 95% confidence interval) of subtypes. ED and brain metastasis recurrence in subtypes are summarized by OR (pink; top axis) and proportions (sky blue; bottom axis). IHC test results of seven markers to evaluate subtype. Circle size indicates the percentage of IHC-positive samples for each marker. Subtype-specific markers (* *p* < 1 × 10^−4^) are shown with an asterisk. (**D**) The GSEA result table was obtained from ReactomeFI using DEGs for each subtype.

**Figure 2 cancers-15-03568-f002:**
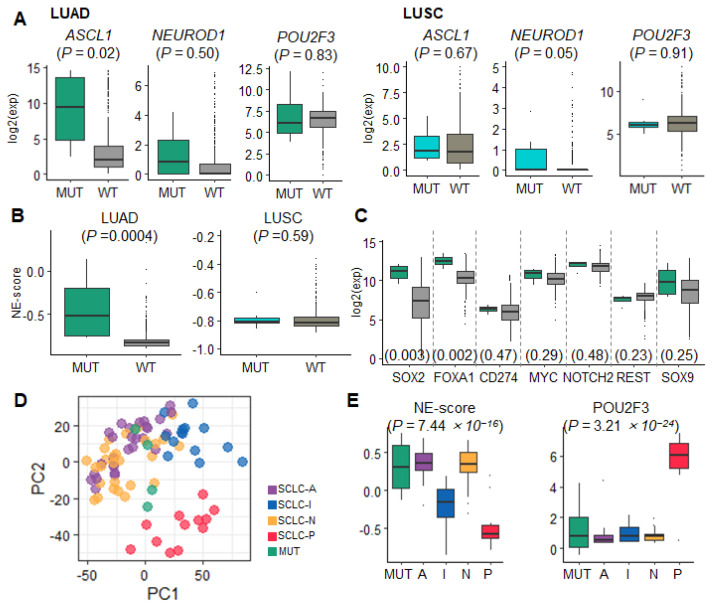
Data distribution according to *TP53/RB1* (MUT) mutation status in the TCGA cohort. *P* values were calculated using the Wilcoxon rank-sum test. (**A**) Box plots illustrating the expression of three SCLC master regulators in LUAD and LUSC samples in the MUT or wild-type (WT) group. (**B**) Box plots of the NE scores of LUAD and LUSC samples for each group. (**C**) Box plots of the expression of different SCLC subtype regulators in LUAD samples for each group. (**D**) A PCA scatter plot of the SCLC and LUAD MUT groups. (**E**) Box plots of NE scores and the expression of the non-NE-type regulator *POU2F3* in the LUAD MUT and SCLC subtypes.

**Figure 3 cancers-15-03568-f003:**
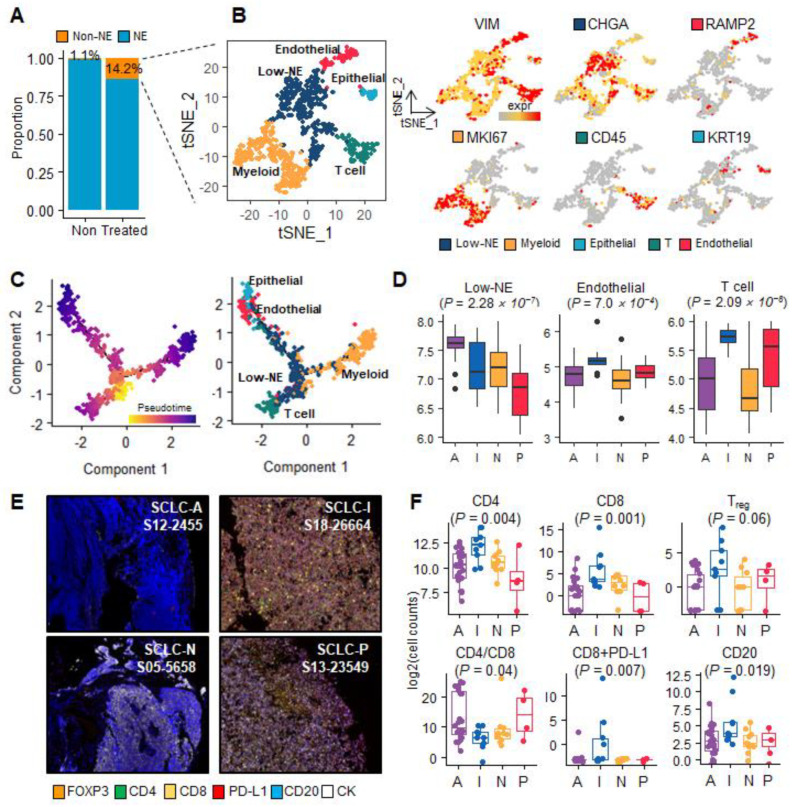
The clonal heterogeneity of SCLC TME cells represents the distinct molecular features of subtypes. (**A**) NE and non-NE cell proportions for the treatment-naïve (non) and treatment (treated) patients. (**B**) t-SNE plots representing non-NE cell clustering to classify five cell types and gene expression representative markers. (**C**) Trajectory tree plots of non-NE cells colorized by pseudotime and five TME cell types. (**D**) Score box plots for each TME subcluster gene signature according to the SCLC subtypes. Low-NE, endothelial, and T-cell types exhibited significant differences among the four subtypes (*p* < 0.0007). (**E**) Representative mIHC/IF images for each SCLC subtype. Six-color, multiplex immunofluorescent images of SCLC tissue sections demonstrating the spatial distribution of different immune lineages and markers within the stroma and tumor regions of each subtype. The following five proteins were labeled with this technique: FOXP3 (orange), CD4 (green), CD8 (yellow), PD-L1 (red), CD20 (blue), and CK (white). (**F**) Box plots of log2-scale cell counts for each marker according to subtype.

**Figure 4 cancers-15-03568-f004:**
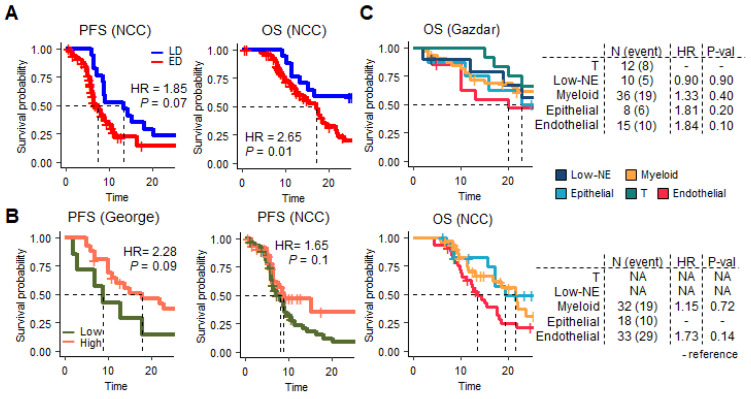
Survival plots of the NCC and George et al.’s cohorts. The P value and hazard ratio (HR) were denoted for each plot using Cox regression analysis. A two-year survival analysis was performed. (**A**) PFS and OS significance in NCC cohort LD (blue) and ED (red) status. (**B**) PFS according to NE score (low: green, high: orange). (**C**) OS plots according to the activation of five TME subclusters in George et al. and NCC cohorts.

**Figure 5 cancers-15-03568-f005:**
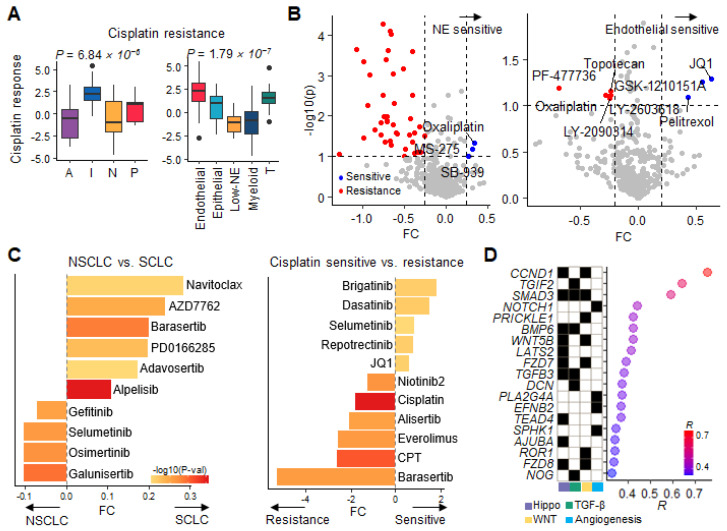
SCLC drug response. (**A**) Boxplots of cisplatin resistance scores with subtypes and TME subcluster types. (**B**) Volcano plots illustrating drug responses according to NE vs. non-NE and endothelial vs. nonendothelial comparisons. FC (*x*-axis) and log-scale *p* value (*y*-axis) were calculated using t-test. Blue and red dots indicate sensitive and resistant drugs, respectively. (**C**) Bar plots of drug response comparing NSCLC vs. SCLC (left) and cisplatin-sensitive vs. cisplatin-resistant SCLC (right) in PDCs. The X-axes are FC for drugs sorted by FC, and the bar color scale represents –log 10 (*p* value). (**D**) A dot plot of correlation coefficients between JQ1 drug response and gene expression. The heatmap denotes pathways in which genes are involved.

## Data Availability

The data can be shared up on request.

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
