# Peer review of "Molecular Subtypes and Tumor Microenvironment Characteristics of Small-Cell Lung Cancer Associated with Platinum-Resistance"

_cancers, 2023, doi:10.3390/cancers15143568_

Round 1

Reviewer 1 Report

The study findings are exciting, and I admire the team's detailed analysis and explanation. However, here are some comments to make the manuscript acceptable to more readers. 

Introduction and Abstract: Please use the common term "tumor microenvironment (TME)" instead of "tumor-associated microenvironment (TME)" overall the manuscript. It would be great to see a section on TME, lung TME, SCLC TME, and common ex-vivo and in vivo models to study SCLC TME. I request write the aims and objectives of the study in a separate paragraph.

Discussion:  At the end of the Discussions, please add a section on strengths with limitations of the study, and another one on future directions.

Author Response

We appreciated review’s interests and comments. All modifications are highlighted to red fond, and we mentioned line number for each answer.

Reviewer 2 Report

Comments

This is a well-written manuscript titled “Molecular subtype of small-cell lung cancer emerged tumor microenvironment characteristics by development and chemotherapy” demonstrated the bromodomain and extra-terminal (BET) inhibitor JQ1 efficacy and sensitivity to SCLC-I in which EndMT signal conferred platinum resistance in order to improve therapeutic strategies. The author further showed that BET inhibitors suppressed the aggressive angiogenesis phenotype of SCLC-I. They have also revealed that    EndMT development contributed to a poor outcome in SCLC-I and heterogenous TME development facilitated platinum resistance. BET inhibitors are novel candidates to overcome platinum resistance. This manuscript will be of good interest to the scientific community. A manuscript can be accepted in its present form with minor revision.

1.      BET inhibitor targeting BRD4, suppresses angiogenesis, it will be interesting to show BET inhibitor suppresses angiogenesis experimentally in SCLC-I cells like angiogenesis Tube Formation Assays, Endothelial Adhesion, Invasion, and Migration Assays Scratch Wound Healing Assays.

2.     There is not any specific readout or data to show inhibition of BRD4 by JQ1. It will be interesting to show the inhibition of acetylation of H3 or H4 histone.

3.     How BET inhibitor targets BRD4 enrichment of the extracellular matrix organization, Hippo, WNT, TGF-β signaling, and angiogenesis pathways. It will be interesting to show some experimental evidence of protein level alteration of the mentioned pathway.

4.     Basically, SCLC is marked by neuroendocrine (NE) and non-neuroendocrine (non-NE) cell states. Further SCLC is defined by differential expression of transcription factors: ASCL1 (SCLC-A), NEUROD1 (SCLC-N), POU2F3 (SCLCP), and YAP1 (SCLC-Y). I am wondering if SCLC-Y also shows the same level of sensitivity against JQ1.   

Dear Editor,

The manuscript is well-written and nicely structured. I thoroughly enjoyed  reading this manuscript and easy to understand. I think the manuscript needs minor proofreading for spelling mistakes before acceptance for publication.

Regards,

Ajit  

Author Response

We appreciated reviewer’s interests and comments. All modifications are highlighted to red fond, and we mentioned line number for each answer.
